# Fibroblast Growth Factor Family in the Progression of Prostate Cancer

**DOI:** 10.3390/jcm8020183

**Published:** 2019-02-04

**Authors:** Jun Teishima, Tetsutaro Hayashi, Hirotaka Nagamatsu, Koichi Shoji, Hiroyuki Shikuma, Ryoken Yamanaka, Yohei Sekino, Keisuke Goto, Shogo Inoue, Akio Matsubara

**Affiliations:** Department of Urology, Graduate School of Biomedical and Health Sciences, Hiroshima University, 1-2-3 Kasumi, Minami-ku, Hiroshima 734-8551, Japan; tetsutaro.hayashi@gmail.com (T.H.), k717171k@yahoo.co.jp (H.N.), urokshoji@yahoo.co.jp (K.S.), himuro.49.1026@gmail.com (H.S.), yamanaka_ryouken@hiro-hosp.jp (R.Y.), akikosekino@gmail.com (Y.S.), keigoto@hiroshima-u.ac.jp (K.G.), inosyogo@hiroshima-u.ac.jp (S.I.), matsua@hiroshima-u.ac.jp (A.M.)

**Keywords:** fibroblast growth factor, fibroblast growth factor receptor, prostate cancer

## Abstract

Fibroblast growth factors (FGFs) and FGF receptors (FGFRs) play an important role in the maintenance of tissue homeostasis and the development and differentiation of prostate tissue through epithelial-stromal interactions. Aberrations of this signaling are linked to the development and progression of prostate cancer (PCa). The FGF family includes two subfamilies, paracrine FGFs and endocrine FGFs. Paracrine FGFs directly bind the extracellular domain of FGFRs and act as a growth factor through the activation of tyrosine kinase signaling. Endocrine FGFs have a low affinity of heparin/heparan sulfate and are easy to circulate in serum. Their biological function is exerted as both a growth factor binding FGFRs with co-receptors and as an endocrine molecule. Many studies have demonstrated the significance of these FGFs and FGFRs in the development and progression of PCa. Herein, we discuss the current knowledge regarding the role of FGFs and FGFRs—including paracrine FGFs, endocrine FGFs, and FGFRs—in the development and progression of PCa, focusing on the representative molecules in each subfamily.

## 1. Introduction

Prostate cancer (PCa) is one of the most common hormone-dependent cancers. Androgen-deprived therapy (ADT) has been the standard option for PCa. It is initially effective in most PCa cases; however, PCa becomes refractory for ADT in spite of the castration level of serum testosterone, which is called “castration-resistant prostate cancer” (CRPC). There have been multiple studies on the efficacy of various agents that include androgen-receptor (AR) targeted agents and anticancer drugs. 

Many studies have demonstrated the aberrant activation of fibroblast growth factor (FGF)/FGF receptor (FGFR) signaling in several cancers, including head and neck, lung, breast, endometrial, bladder, and prostate cancer [1,2]. Herein, we review the FGF family’s involvement in the development and progression of prostate cancer. We mainly discuss the representative molecules in each subfamily: FGF9 as classic FGFs, FGF19 as endocrine FGFs, and FGFR2IIIb. 

## 2. FGFs and FGFRs

The human *Fgf* gene family consists of 22 members, and they are classified into seven subfamilies based on phylogenetic analysis (Figure 1) [3]. FGFs are also classified into three subfamilies (intracrine, paracrine, and endocrine FGFs) based on their mechanism of action. Intracrine FGFs are not typical and does not bind FGFR. Their function is not mediated by a receptor. Therefore, we herein focused two subfamilies, paracrine FGFs and endocrine FGFs. Paracrine FGFs—which consist of 15 members—exert their biological function through binding to an extracellular domain of FGFRs with heparin/ heparan sulfate and activating tyrosine kinase signaling of FGFRs. 

Endocrine FGFs consist of FGF19, FGF21, and FGF23. These molecules have a low affinity to heparin/heparan sulfate in contrast to paracrine FGFs. They have to form complexes with co-receptors, α/β-Klotho, to bind to the extracellular domain of FGFR. Because the endocrine FGFs’ affinity to FGFRs changes depending on the existence of α/β-Klotho, they have hormone-like activity beyond functioning as a growth factor (Figure 2). Several studies have demonstrated the physiological function of endocrine FGFs (Table 1) [4]. Among endocrine FGFs, FGF15 (the mouse orthologue of human FGF19) was the first molecule to be identified. The physiological activity of FGF19, including the regulation of glucose and bile acid metabolism, is exerted through the formation of a complex with FGFR4 and beta-klotho and follows the activation of a signaling cascade by recruiting adaptor proteins in cytoplasm [5,6,7]. FGF21 mainly acts as a metabolic regulator in the liver, adipose tissue, and the pancreas. The tissue-specific metabolic action of FGF21 depends on its specificity to the receptor. FGF21 binds with beta-klotho and FGFR1c [6,8,9]. FGF23 is a bone-derived endocrine hormone. Expression of FGF23 is induced by activation of the vitamin D receptor (VDR) with 1,25 dihydroxyvitamin D (1,25D), and FGF23 is a suppressor of 1,25D. So FGF23 and 1,25D are linked by mutual regulation. A signaling cascade of FGF23 activates through formation of a complex with alfa-klotho and FGFR1 for the kidney and FGFR3c for the parathyroid gland. FGF23 regulates phosphate and vitamin D metabolism in the kidney and inhibits parathyroid hormone secretion and vitamin D synthesis in the parathyroid gland [10,11].

Four FGFRs, FGFR1-4, contain an extracellular ligand binding domain with three immunoglobulin (Ig)-like domains (I–III), a transmembrane domain, and a split intracellular tyrosine kinase domain. FGFR1-3 have two kinds of Ig-like III domains, IIIb and IIIc, which are generated by alternative splicing. The Ig-like domain is crucial for determining ligand-binding specificity, and as a result, seven FGFR with different ligand-binding specificities are derived from four *Fgfr* genes [12]. 

## 3. FGF-FGFR Signaling in Epithelial-Stromal Interaction in Prostate Tissue

Epithelial-stromal interaction plays an important role in maintaining the homeostasis in normal prostatic tissue [13,14]. Stromal tissues secrete paracrine factors that include FGF ligands, and they lead to stimulation of epithelial maintenance and growth. Huang et al. reported the significance of FGFR2 signaling for preserving stemness and preventing differentiation of prostate stem cells [15]. FGFR2IIIb, a splicing variant of the FGFR2, is a resident form of FGFRs expressed in normal prostate epithelial cells. FGFR2IIIb is specific to FGF7, and FGF7-FGFR2IIIb contributes epithelial-stromal interaction [16]. The loss of FGFR2 isoforms is shown in human PCa tissues, and the loss of FGFR2IIIb is associated with the characteristics of castration-resistant prostate cancer (CRPC) in particular [17]. 

The expression pattern of FGFR is different in each PCa cell line. For instance, FGFR1 expression is detectable and FGFR2IIIb expression is undetectable in PC3 cells that show androgen-independent growth and high potential of cell proliferation. On the other hand, FGFR2IIIb expression is detectable in LNCap cells that show expression of androgen receptor (AR), androgen-dependent growth, and relatively low potential of cell proliferation [18]. In addition to FGFs and FGFRs, FGF receptor substrate 2alpha (FRS2alpha), an FGFR interactive adaptor protein, involves multiple signaling pathways to the activated FGFR kinase. FRS2 alfa is required for prostate development and tumorigenesis [19], as well as in angiogenesis [20].

## 4. Effects of the Restoration of FGFR2IIIb in Prostate Cancer Cells

Many studies have demonstrated the association of aberrant FGFR signaling with the development and progression of PCa [21,22]. Binding ligands, FGFRs form functional dimerization and lead transphosphorylation and activation of downstream signaling pathways such as Ras, Src, PKCγ, MAPK, PI3K-AKT, and STAT [23,24,25]. The involvement of FGF signaling in various molecular mechanisms has been reported in PCa. Shao et al. reported that FGF-FGFR signaling plays an important role in transformation induced by the loss of a PTEN tumor suppressor when combined with the expression of the TMPRSS2/ERG fusion gene [26], and activation of FGF-FGFR signaling by FGF8b overexpression in PTEN deficiency is reported to be associated with prostate tumorigenesis [27]. FGF-FGFR signaling is also related to the induction of an inflammatory response in PCa tissues [28]. The involvement of aberrant FGFR1 signaling in the progression of PCa in particular was demonstrated in several studies. FGFR1 signaling promotes the reprogramming of energy metabolism from oxidative phosphorylation to aerobic glycolysis by regulating the expression of an LDH isoenzyme [29]. It also promotes an inflammatory response through activation of NF-κB signaling [30]. Furthermore, activation of FGFR1 signaling promotes epithelial to mesenchymal transition and androgen independency in PCa cells [31,32]. Loss of FGFR2IIIb and enhancement of ectopic expression of FGFR1 PCa progression have been reported as common events in the progression of PCa [21]. As FGFR2IIIb plays an important role in the maintenance and its disorder is found in PCa cell lines and tissues, several investigators have reported the effects of restoring FGFR2IIIb. In animal models and PCa cell lines, FGFR2IIIb’s restoration also restored responsiveness to stroma and significantly reduced in vivo tumorigenesis. In castration-resistant human PCa cells, restoration of FGFR2IIIb showed the inhibition of cell proliferation, the induction of differentiation, and the enhancement of apoptosis in a ligand-independent manner [21,33,34]. In addition, in PCa cells overexpressing FGFR2IIIb, clonogenic cell death increased in concurrence with enhanced apoptosis and cell cycle arrest in the G2/M-phase and radiosensitivity by gamma-irradiation [35]. Another study reported the effect FGFR2IIIb’s restoration had on the chemosensitivity in PCa cells. Restoration of FGFR2IIIb led to the enhanced chemosensitivity of several agents, especially docetaxel. The expression of N-cadherin, vimentin, survivin, and XIAP were induced by restoring FGFR2IIIb [36]. This data indicates that PCa cell lines are induced to a more differentiated phenotype when changing the pattern of gene expression that became sensitive to radiation and chemotherapy when FGFR2IIIb was restored.

## 5. Involvement of FGFs in the Development and Progression of Prostate Cancer

### 5.1. Paracrine FGFs

Upregulated expression of FGF1, FGF2, FGF8, FGF9, and FGF10 were shown in human PCa [37]. Murine studies demonstrated the epithelial and mesenchymal interaction using a FGF/FGFR complex in PCa [38]. Pecqueux et al. demonstrated the association between strong expression of FGF2 in tumor stroma and a high postoperative recurrence rate and that exogenous FGF2 can drive genomic instability to promote PCa progression through enhancement of DNA damage [39]. Cuevas et al. reported the linkage between altered micro-environmental signaling that includes FGF2 overexpression and mitotic instability [40]. In the study focused on a bone metastatic site, FGF2 was upregulated in osteoblast and promoted the proliferation of PCa cells under the loss of TGFβ signaling [41]. These reports indicated the significance of FGF/FGFR signaling through paracrine FGFs (especially FGF2) in a cancer micro-environment for the development and progression of PCa. 

FGF9 is an abundant molecule in nervous tissue and soft tissue, while it has been reported to be a key molecule of epithelial-stromal interaction [42]. Several studies have demonstrated that FGF9 was associated with the proliferation of glia [43], regulation of the differentiation of astrocytes [44] and oligodendrocytes [45], and the regulation of joint formations [46,47]. FGF9’s involvement in malignant neoplasms has been reported in glioma [48], ovarian cancer [49], and lung cancer [50]. In prostate epithelial cells, overexpression of FGF9 lead the augmentation of reactive stroma formation and promoted initiation and progression of PCa [51]. In our study, cell viability and invasion of LNCaP was significantly enhanced, and expression of MMP2, N-cadherin, and VEGF-A were induced in LNCaP incubated in medium with FGF9. In immuno-histochemical staining, the prevalence of both VEGF-A and N-cadherin-positive cells was significantly higher compared to FGF9-negative cases [52]. The biochemical relapse-free survival (bRFS) rate in cases with FGF9-positive cases was significantly lower than that in FGF9-negative cases [53]. “FGF9-positive” in this study was determined based on the findings of immunohistochemistry staining on just one representative section. And even in FGF9-positive cases, just a small population of FGF9-positive cells with very aggressive pathological features was present. In other words, FGF9 was only positive in especially high-grade cancer cells in cases with localized PCa. Furthermore, several studies have reported FGF9 in PCa at an advanced stage. Accumulation of FGF9 to the region of bone metastasis formed by AR-negative PCa cells indicated that FGF9 was a key factor in formation of bone metastasis [54]. In AR-negative CRPC cases, neuroendocrine differentiation (NE) is one of the most representative phenotypes. In a further dedifferentiated phenotype, “AR-negative and NE-negative”, expression of FGF1, FGF8, and FGF9 increased, and MAPK pathway was activated [55]. The results of these studies indicated that FGF9 might be a key molecule for an advanced PCa that include CRPC rather than a localized one.

### 5.2. Endocrine FGFs

Endocrine FGFs have two different characteristics, a growth factor and a metabolic regulator. Investigating endocrine FGFs might clarify the molecular link between the progression of cancer and metabolism. Among endocrine FGFs, the association between FGF19 and malignant diseases such as liver cancer, colon cancer, and prostate cancer was demonstrated [56,57,58,59]. FGF19 induces the expression of markers of epithelial mesenchymal transition in hormone-sensitive prostate cancer cells [60,61]. Consistent with the results, accumulation of FGF19 in cytoplasm was shown in poorly differentiated prostate cancer cells in human prostate cancer tissues derived from radical prostatectomy, and the presence of FGF19-positive tissues correlated with positive immuno-histochemical staining with N-cadherin in prostate cancer tissues [60]. The bRFS rate in cases with FGF19-positive tissues was significantly lower than in cases with FGF19-negative tissues [60]. In prostate cancer cells, FGF19 stimulates cell proliferation and cell invasion through activation of MAP kinase and AKT pathways [62]. Expression of FGF19 increased in castration-resistant cell lines compared with castration-sensitive ones or immortalized normal prostate cells [62]. According to these reports, FGF19 might be associated with the risk of post-operative recurrence by enhancement of cell proliferation and epithelial-mesenchymal transition of PCa cells. 

Since endocrine FGFs that include FGF19 act as circulating hormones related to several metabolic diseases, the impact of their serum concentration for metabolic diseases was investigated. These molecules are expected to be a potential serum biomarker for PCa. We measured serum FGF19 and beta-klotho level in cases with PCa. While there was no relationship between the serum klotho level and pathological findings, the results showed that patients with a high Gleason score had higher serum FGF19 levels than those with a low Gleason score [60]. This data indicates that FGF19 might be a potential serum biomarker in PCa. One limitation to using endocrine FGFs that include FGF19 as a serum biomarker is the change of serum concentration according to dietary conditions and blood sugar level because endocrine FGFs act as a metabolic regulator and their level is also regulated by a feedback mechanism. Further study will clarify the optimal timing and conditions for measuring endocrine FGFs to apply them as a serum biomarker for PCa. Besides FGF19, the increased expression of FGF23 in many PCa cell lines and PCa tissues was reported. FGF23 enhances proliferation, invasion through activation of AKT, and the MAPK pathway in PCa cell lines. These findings indicate that it can promote PCa progression in an autocrine, paracrine, and/or endocrine manner [63].

## 6. Conclusions

In this article, we described the impact of FGF-FGFR abnormalities on the development and/or the progression of PCa. Since we found the heterogeneity of PCa, it is important to clarify and understand various molecular mechanisms of PCa in order to determine the most appropriate therapeutic strategies.

## Figures and Tables

**Figure 1 jcm-08-00183-f001:**
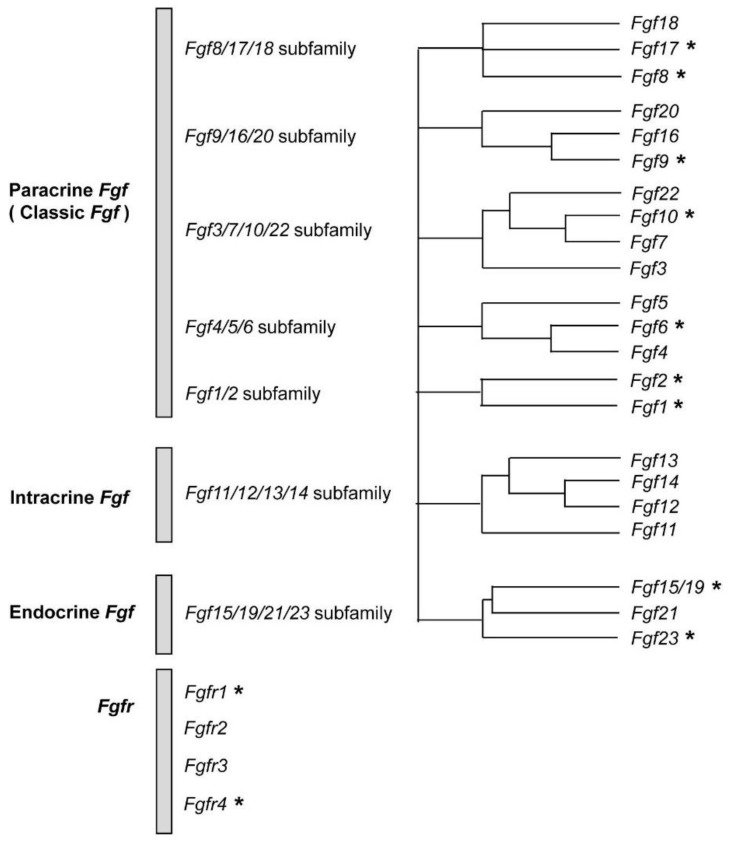
*Fgf* genes consisting of 7 subfamilies and *Fgfr* genes. Asterisks indicates fibroblast growth factor (FGF)/ fibroblast growth factor receptor (FGFR) whose expression are enhanced in prostate cancer cells and/or tissues.

**Figure 2 jcm-08-00183-f002:**
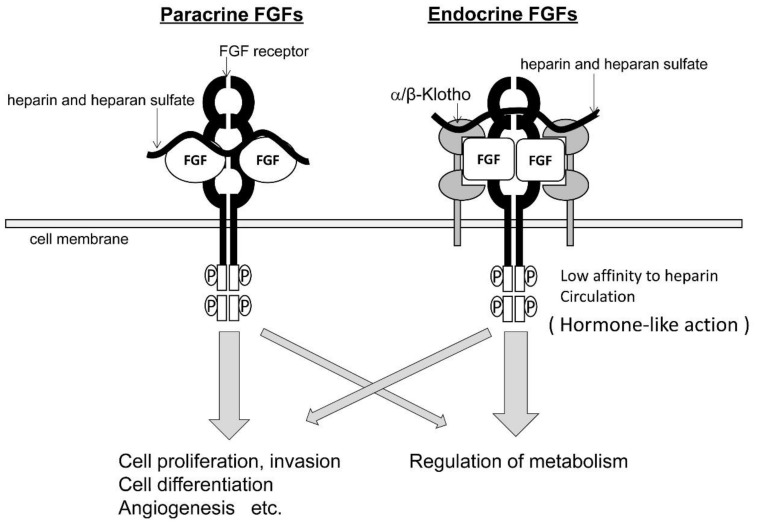
Difference in the mechanism in signal transduction between paracrine FGFs and endocrine FGFs.

**Table 1 jcm-08-00183-t001:** Pathophysiological activity of endocrine FGFs.

	Up-Regulation	Down-Regulation	Increase	Decrease
FGF19				
	Glycogen synthesis	Bile acid synthesis	Extrahepatic cholestasis	IBD
		Gluconeogenesis	Chronic hemodialysis	NAFLD
				Primary bile acid malabsorption
				Obesity
FGF21				
	Hepatic fatty acid oxidation	Ovulation	Type 2 diabetes	Anorexia
	Ketogenesis	Growth hormone signaling	Metabolic syndrome	Nervosa
	Glucogenesis		NAFLD	
	Thermogenesis		Coronary heart disease	
	WAT browing			
	Growth hormone resistance			
	Weight loss			
	Ovulation			
FGF23				
	Calcium secretion	Renal phosphate absorption	ADHR	Hemodialysis
	Life span	Bone and renal calcium reabsorption	XLH rickets	Familial tumoral calcinosis
		Vitamin D synthesis	TIO	
		PTH secretion	Cardiac hypertrophy	

ADHR, autosomal dominant hypophosphataemic rickets; FGF, fibroblast growth factor; IBD, irritable bowel disease; NAFLD, non-alcoholic fatty liver disease; PTH, parathyroid hormone; TIO, tumor-induced osteomalacia; WAT, white adipose tissue; and XLH, X-linked hypophosphataemic.

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
