# Peer review of "Fibroblast Growth Factor Family in the Progression of Prostate Cancer"

_jcm, 2019, doi:10.3390/jcm8020183_

Reviewer 1 Report

The authors have reviewed the role of fibroblast growth factor family in prostate cancer. It is an interesting article but some improvementes should be carried out before acceptance.

1- The table 1 shows the effects of FGFs in physiology and pathology. The legend is more correct "Pathophysiological activity of endocrine FGFs".

2- A subhead should be inserted about the expression of FGFRs expression in prostate, highlighting the differentially expressed between normal and prostate cancer. A table should be inserted summarizing their expression.

3- A table summarizing the expression of FGFs should be inserted.

Author Response

Dear Editor,

We are most grateful for you to review our manuscript entitled No. jcm-434016-v2 entitled “Fibroblast growth factor family in the progression of prostate cancer” for your consideration for publication in the Special Issue in Journal of Clinical Medicine”, entitled “Cytobiology of Human Prostate Cancer Cells and Its Clinical Applications”.

Based on the comments from the editor and reviewers, we have improved our manuscript and revised.

All our comments and correction in this manuscript have been written by red color.

Reviewer 1:

The authors have reviewed the role of fibroblast growth factor family in prostate cancer. It is an interesting article but some improvementes should be carried out before acceptance.

1-The table 1 shows the effects of FGFs in physiology and pathology. The legend is more correct "Pathophysiological activity of endocrine FGFs".

Authors’ reply to question 1) Thank you for the comment. The legend has been corrected based on the suggestion.

2- A subhead should be inserted about the expression of FGFRs expression in prostate, highlighting the differentially expressed between normal and prostate cancer. A table should be inserted summarizing their expression.

3- A table summarizing the expression of FGFs should be inserted.

Authors’ reply to question 2 and 3) Thank you for the comment. I agree with reviewer’s suggestion. Expression of FGF and FGFR in prostate is an important information. Expression of many FGFs and FGFR1 and 4 are enhanced in PCa cells and tissues compared with normal prostate tissue. Instead of Tables, FGFs/FGFRs whose expression are increased in PCa have been marked by asterisks in Figure 1.

Reviewer 2 Report

The review summarizes the role of fgf signaling in the prostate. however, the authors left out several reports, such as frs2a is required for prostate development and tumorigenesis (development), as well as in angiogenesis (oncogene) fgf9 promotes reactive stromal (international j of biosciences). it would also help reader to understand fgf signaling in prostate stem cells if the author can also cite fgfr2 in prostate stem cells (JBC).

in the abstract, the authors stated "The FGF family includes two subfamilies, paracrine FGFs 14 and endocrine FGFs", however, in the text, the author also mentioned "intercrine" fgf. since fgf11-14 are not typical fgf, and did not bind to fgfr. their functions are not mediated by a receptor, i would suggest the authors do not use "intercrine"

 Author Response

Dear Editor,

We are most grateful for you to review our manuscript entitled No. jcm-434016-v2 entitled “Fibroblast growth factor family in the progression of prostate cancer” for your consideration for publication in the Special Issue in Journal of Clinical Medicine”, entitled “Cytobiology of Human Prostate Cancer Cells and Its Clinical Applications”.

Based on the comments from the editor and reviewers, we have improved our manuscript and revised.

All our comments and correction in this manuscript have been written by red color.

Reviewer 2:

The review summarizes the role of fgf signaling in the prostate. however, the authors left out several reports, such as frs2a is required for prostate development and tumorigenesis (development), as well as in angiogenesis (oncogene) fgf9 promotes reactive stromal (international j of biosciences). it would also help reader to understand fgf signaling in prostate stem cells if the author can also cite fgfr2 in prostate stem cells (JBC).

Authors’ reply) Thank you for the comment. All papers suggested by reviewers have been cited and the description about frs2a, fgf9 and fgfr2 has been inserted based on the suggestion.

in the abstract, the authors stated "The FGF family includes two subfamilies, paracrine FGFs 14 and endocrine FGFs", however, in the text, the author also mentioned "intercrine" fgf. since fgf11-14 are not typical fgf, and did not bind to fgfr. their functions are not mediated by a receptor, i would suggest the authors do not use "intercrine"

Authors’ reply) Thank you for the comment. I agree with the comment that fgf11-14 are not typical. Most previous studies focused just paracrine FGFs and endocrine FGFs. The statement about intracrine FGFs has been deleted based on the suggestion.